# Signatures of a time-reversal symmetric Weyl semimetal with only four Weyl points

Ilya Belopolski[1], Peng Yu[2], Daniel S. Sanchez[1], Yukiaki Ishida[3], Tay-Rong Chang[4,5], Songtian S. Zhang[1], Su-Yang Xu[1], Hao Zheng [1], Guoqing Chang [6,7], Guang Bian[1,8], Horng-Tay Jeng[4,9], Takeshi Kondo[3], Hsin Lin[6,7], Zheng Liu [2,10,11], Shik Shin[3] & M. Zahid Hasan [1,12]

Through intense research on Weyl semimetals during the past few years, we have come to appreciate that typical Weyl semimetals host many Weyl points. Nonetheless, the minimum nonzero number of Weyl points allowed in a time-reversal invariant Weyl semimetal is four. Realizing such a system is of fundamental interest and may simplify transport experiments. Recently, it was predicted that $TaIrTe_4$ realizes a minimal Weyl semimetal. However, the Weyl points and Fermi arcs live entirely above the Fermi level, making them inaccessible to conventional angle-resolved photoemission spectroscopy (ARPES). Here, we use pump-probe ARPES to directly access the band structure above the Fermi level in $TaIrTe_4$. We observe signatures of Weyl points and topological Fermi arcs. Combined with ab initio calculation, our results show that $TaIrTe_4$ is a Weyl semimetal with the minimum number of four Weyl points. Our work provides a simpler platform for accessing exotic transport phenomena arising in Weyl semimetals.

---

[1] Laboratory for Topological Quantum Matter and Spectroscopy (B7), Department of Physics, Princeton University, Princeton, NJ 08544, USA. [2] Centre for Programmable Materials, School of Materials Science and Engineering, Nanyang Technological University, Singapore 639798, Singapore. [3] Institute for Solid State Physics (ISSP), University of Tokyo, Kashiwa-no-ha, Kashiwa, Chiba 277-8581, Japan. [4] Department of Physics, National Tsing Hua University, Hsinchu 30013, Taiwan. [5] Department of Physics, National Cheng Kung University, Tainan 701, Taiwan. [6] Centre for Advanced 2D Materials and Graphene Research Centre, National University of Singapore, 6 Science Drive 2, Singapore 117546, Singapore. [7] Department of Physics, National University of Singapore, 2 Science Drive 3, Singapore 117542, Singapore. [8] Department of Physics and Astronomy, University of Missouri, Columbia, MO 65211, USA. [9] Institute of Physics, Academia Sinica, Taipei 11529, Taiwan. [10] NOVITAS, Nanoelectronics Centre of Excellence, School of Electrical and Electronic Engineering, Nanyang Technological University, Singapore 639798, Singapore. [11] CINTRA CNRS/NTU/THALES, UMI 3288, Research Techno Plaza, 50 Nanyang Drive, Border X Block, Level 6, Singapore 637553, Singapore. [12] Princeton Institute for Science and Technology of Materials, Princeton University, Princeton, NJ 08544, USA. Ilya Belopolski and Peng Yu contributed equally to this work. Correspondence and requests for materials should be addressed to I.B. (email: ilyab@princeton.edu) or to M.Z.H. (email: mzhasan@princeton.edu)

A Weyl semimetal is a crystal which hosts emergent Weyl fermions as electronic quasiparticles. In an electronic band structure, these Weyl fermions correspond to accidental degeneracies, or Weyl points, between two bands[1–5]. It is well-understood that Weyl points can only arise if a material breaks either spatial inversion symmetry, $\mathcal{I}$, or time-reversal symmetry, $\mathcal{T}$[6–9]. At the same time, in a Weyl semimetal, symmetries of the system tend to produce copies of Weyl points in the Brillouin zone. As a result, typical Weyl semimetals host a proliferation of Weyl points. For instance, the first Weyl semimetals observed in experiment, TaAs and its isoelectronic cousins, have an $\mathcal{I}$ breaking crystal structure, which gives rise to a band structure hosting 24 Weyl points distributed throughout the bulk Brillouin zone[10–17]. However, most of these Weyl points can be related to one another by the remaining symmetries of TaAs, namely two mirror symmetries, $C_4$ rotation symmetry and $\mathcal{T}$. In the $Mo_xW_{1-x}Te_2$ series, which has recently been under intensive theoretical and experimental study as a Weyl semimetal with strongly Lorentz-violating, or Type II, Weyl fermions, mirror symmetry and $\mathcal{T}$ relate subsets of the eight Weyl points[18–25]. As another example, according to calculation, the Weyl semimetal candidate $SrSi_2$ hosts no fewer than 108 Weyl points, copied in sets of 18 by three $C_4$ rotation symmetries[26]. However, as we review below, it is well-known that the minimal nonzero number of Weyl points allowed is 4 for a $\mathcal{T}$ invariant Weyl semimetal. Realizing such a minimal Weyl semimetal is not only of fundamental interest, but is also practically important, because a system

with fewer Weyl points may exhibit simpler properties in transport and be more suitable for device applications.

Recently, $TaIrTe_4$ was predicted to be a Weyl semimetal with only four Weyl points[27]. It was further noted that the Weyl points are associated with Type II Weyl fermions, providing only the second example of a Type II Weyl semimetal after the $Mo_xW_{1-x}Te_2$ series[18]. Moreover, the Weyl points are well-separated in momentum space, with substantially larger topological Fermi arcs as a fraction of the size of the surface Brillouin zone than other known Weyl semimetals. Lastly, $TaIrTe_4$ has a layered crystal structure, which may make it easier to carry out transport experiments and develop device applications. All of these desirable properties have motivated considerable research interest in $TaIrTe_4$. At the same time, one crucial challenge is that the Weyl points and topological Fermi arcs are predicted to live entirely above the Fermi level in $TaIrTe_4$, so that they are inaccessible to conventional angle-resolved photoemission spectroscopy (ARPES).

Here, we observe signatures of Weyl points and topological Fermi arcs in $TaIrTe_4$, realizing the first minimal $\mathcal{T}$ invariant Weyl semimetal. We first briefly reiterate a well-known theoretical argument that the minimum number of Weyl points for a $\mathcal{T}$ invariant Weyl semimetal is four. Then, we present ab initio calculations showing a nearly ideal configuration of Weyl points and Fermi arcs in $TaIrTe_4$. Next, we use pump-probe ARPES to directly access the band structure of $TaIrTe_4$ above the Fermi level in experiment. We report the observation of signatures of

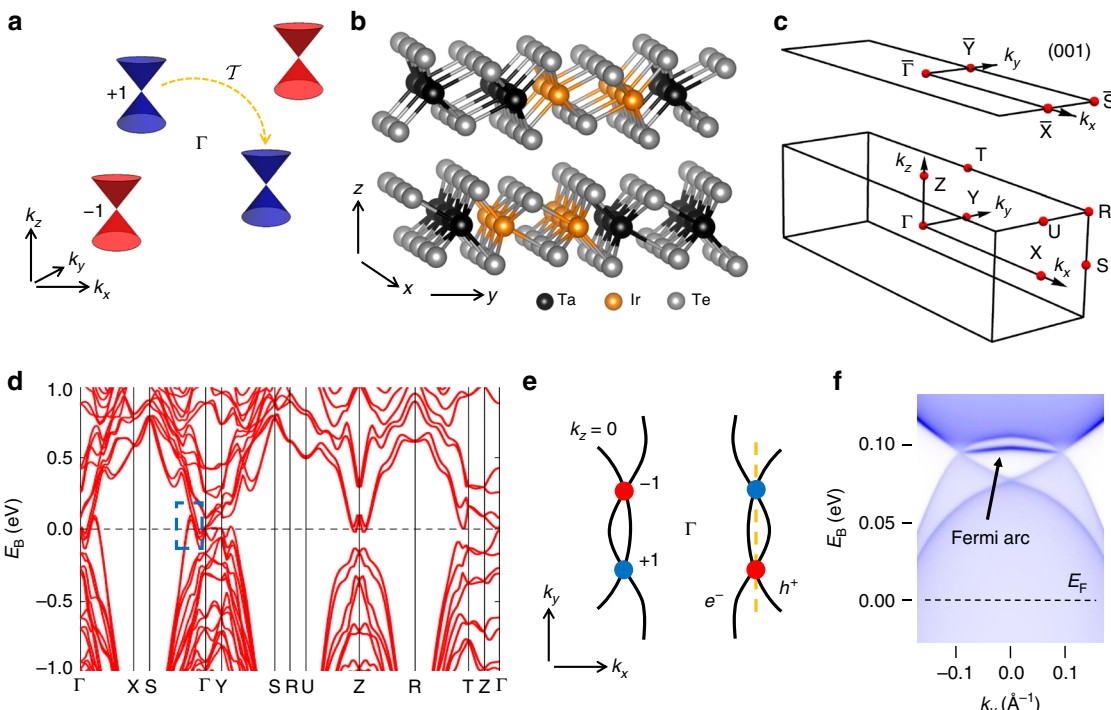

**Fig. 1** Constraints on Weyl points in $\mathcal{T}$ symmetric systems. **a** Illustration of the minimal number of Weyl points in a $\mathcal{T}$ invariant Weyl semimetal. The *blue* and *red* circles and cones represent Weyl points and Weyl cones with ±1 chiral charge at generic *k*-points. In a $\mathcal{T}$ invariant Weyl semimetal, the minimal number of Weyl points is four because $\mathcal{T}$ symmetry sends a Weyl point of a given chiral charge at *k* to a Weyl point of the same chiral charge at −*k* (*orange arrow*). To preserve net zero chiral charge, four Weyl points are needed. **b** The crystal structure of $TaIrTe_4$ is layered, in space group 31, which breaks inversion symmetry. **c** The bulk Brillouin zone (*BZ*) and (001) surface BZ of $TaIrTe_4$ with high-symmetry points marked in *red*. **d** The electronic band structure of $TaIrTe_4$ along high-symmetry lines. There is a band crossing in the region near $\Gamma$, with Weyl points off $\Gamma$ − S (*blue box*). **e** Cartoon illustration of the constant-energy contour at $E_B = E_W$ and $k_z = 0$, with bulk electron and hole pockets which intersect to form Weyl points. A detailed calculation shows that there are in total four Type II Weyl points (*blue* and *red circles*)[27]. **f** Energy-dispersion calculation along a pair of Weyl points in the $k_y$ direction, marked by the *orange line* in **e**. The Weyl points and Fermi arcs live at ~0.1 eV above $E_F$, requiring the use of pump-probe ARPES to directly access the unoccupied band structure to demonstrate a Weyl semimetal

Weyl points and topological Fermi arcs. Combined with ab initio calculations, our results demonstrate that $TaIrTe_4$ has four Weyl points. We conclude that $TaIrTe_4$ can be viewed as a minimal Weyl semimetal, with the simplest configuration of Weyl points allowed in a $\mathcal{T}$ invariant crystal.

## Results

**Minimum number of Weyl points under time-reversal symmetry.** We first reiterate well-known arguments that four is the minimum number of Weyl points allowed in a $\mathcal{T}$ invariant Weyl semimetal. A Weyl point is associated with a chiral charge, directly related to the chirality of the associated emergent Weyl fermion. It can be shown that for any given band the sum of all chiral charges in the Brillouin zone is zero. Further, under $\mathcal{T}$ a Weyl point of a given chiral charge at $k$ is mapped to another Weyl point of the same chiral charge at $-k$. This operation of $\mathcal{T}$ on a chiral charge is illustrated in Fig. 1a on the blue Weyl points with +1 chiral charge (the same arrow applies for the red Weyl points but is not drawn explicitly). Now, if an $\mathcal{I}$ breaking Weyl semimetal has no additional symmetries which produce copies of Weyl points, then the minimum number of Weyl points is fixed by $\mathcal{T}$ symmetry and the requirement that total chiral charge vanish. In the simplest case, $\mathcal{T}$ will produce two copies of Weyl points of chiral charge +1, as shown in Fig. 1a. To balance these out, the system must have two chiral charges of −1, also related by $\mathcal{T}$. In this way, the minimum number of Weyl points in a $\mathcal{T}$ invariant Weyl semimetal is four. This simple scenario is realized in $TaIrTe_4$. The crystal structure of $TaIrTe_4$ is described by space group 31 ($Pmn2_1$), lattice constants $a = 3.77$ Å, $b = 12.421$ Å, and $c = 13.184$ Å, with layered crystal structure, see Fig. 1b. We note that $TaIrTe_4$ takes the same space group as $Mo_xW_{1-x}Te_2$, but has a unit cell doubled along $b$. To study where the Weyl points show up in $TaIrTe_4$ we present the electronic band structure along various high-symmetry directions, see Brillouin zone and ab initio calculation in Fig. 1c, d. Enclosed by the rectangular box along $\Gamma - S$ is a crossing region between the bulk conduction and valence bands that gives rise to Weyl points. A more detailed calculation shows that the Weyl points have tilted over, or Type II, Weyl cones and that they live above the Fermi level, $E_F$ at $k_z = 0$[27]. A cartoon schematic of the resulting constant energy contour at the energy of the Weyl points, $E_B = E_W$, is shown in Fig. 1e. The electron and hole pockets form Type II Weyl cones where they touch (red and blue marks). In this way, $TaIrTe_4$ has four Weyl points, the minimal number allowed in an $\mathcal{I}$ breaking Weyl semimetal. The overall electronic structure of $TaIrTe_4$ near $E_F$ is similar to $Mo_xW_{1-x}Te_2$, but we note that the role of the electron and hole pockets is reversed in $TaIrTe_4$ relative to $Mo_xW_{1-x}Te_2$. Also, $Mo_xW_{1-x}Te_2$ has eight Weyl points, so it is not minimal, and we will see that $TaIrTe_4$ also hosts larger Fermi arc surface states than $Mo_xW_{1-x}Te_2$. To study the expected Fermi arcs in $TaIrTe_4$, we present an energy-dispersion cut along a pair of projected Weyl points along $k_y$, Fig. 1f. We clearly observe a large single Fermi arc surface state at ~0.1 eV above the Fermi level that is ~0.25 Å$^{-1}$ long and connecting a pair of ±1 chiral charged Weyl points along $k_y$. In this way, $TaIrTe_4$ provides a minimal Weyl semimetal with large Fermi arcs. Like the Weyl points, the Fermi arcs live well above the Fermi level, making them inaccessible to conventional ARPES.

**Unoccupied band structure of $TaIrTe_4$ by pump-probe ARPES.** Next, we use pump-probe ARPES to directly access the unoccupied band structure of $TaIrTe_4$ up to $E_B > 0.2$ eV and we find excellent agreement with calculation. In our experiment, we use a 1.48 eV pump laser pulse to excite electrons into low-lying states

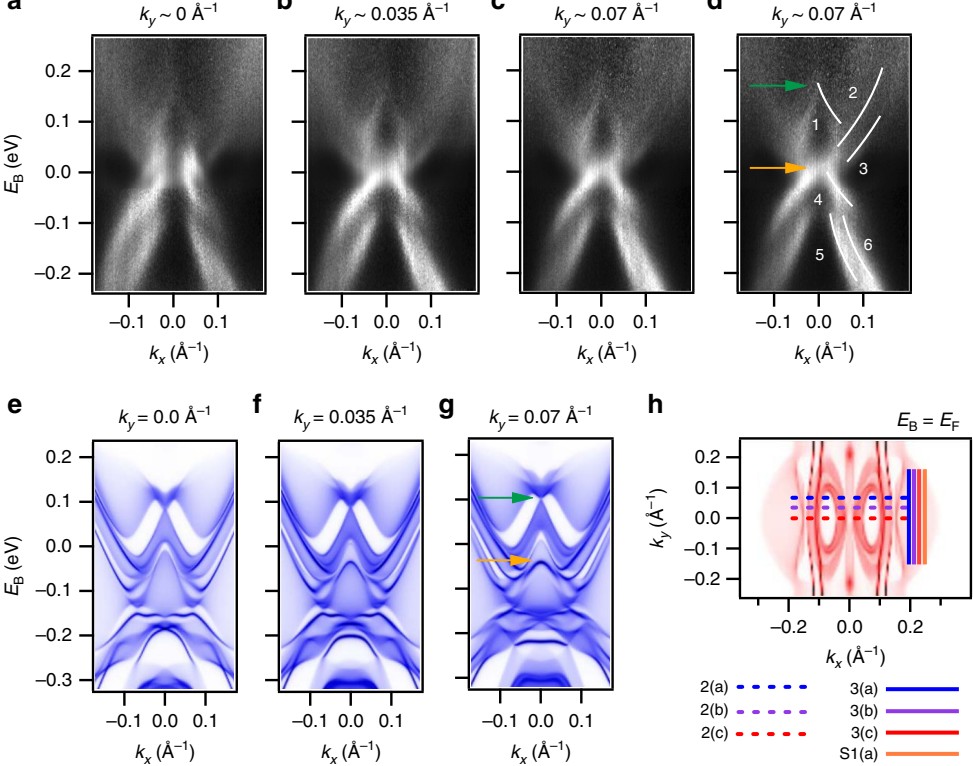

**Fig. 2** Unoccupied electronic structure of $TaIrTe_4$. **a–c** Pump-probe ARPES dispersion maps of $TaIrTe_4$, showing dispersion above $E_F$ at fixed $k_y$ near $\overline{\Gamma}$. **d** Same as **c** but with key features marked. **e, f** Ab initio calculation of $TaIrTe_4$. The data is captured well by calculation, but the sample appears to be hole doped by ~50 meV, comparing the *green* and *orange* arrows in **d**, **g**. **h** Calculation of the nominal Fermi surface, showing weak dispersion along $k_y$ near $\overline{\Gamma}$, consistent with the data. All cuts in Figs. 2 and 3 and Supplementary Fig. 1 are marked (*solid* and *dashed lines*)

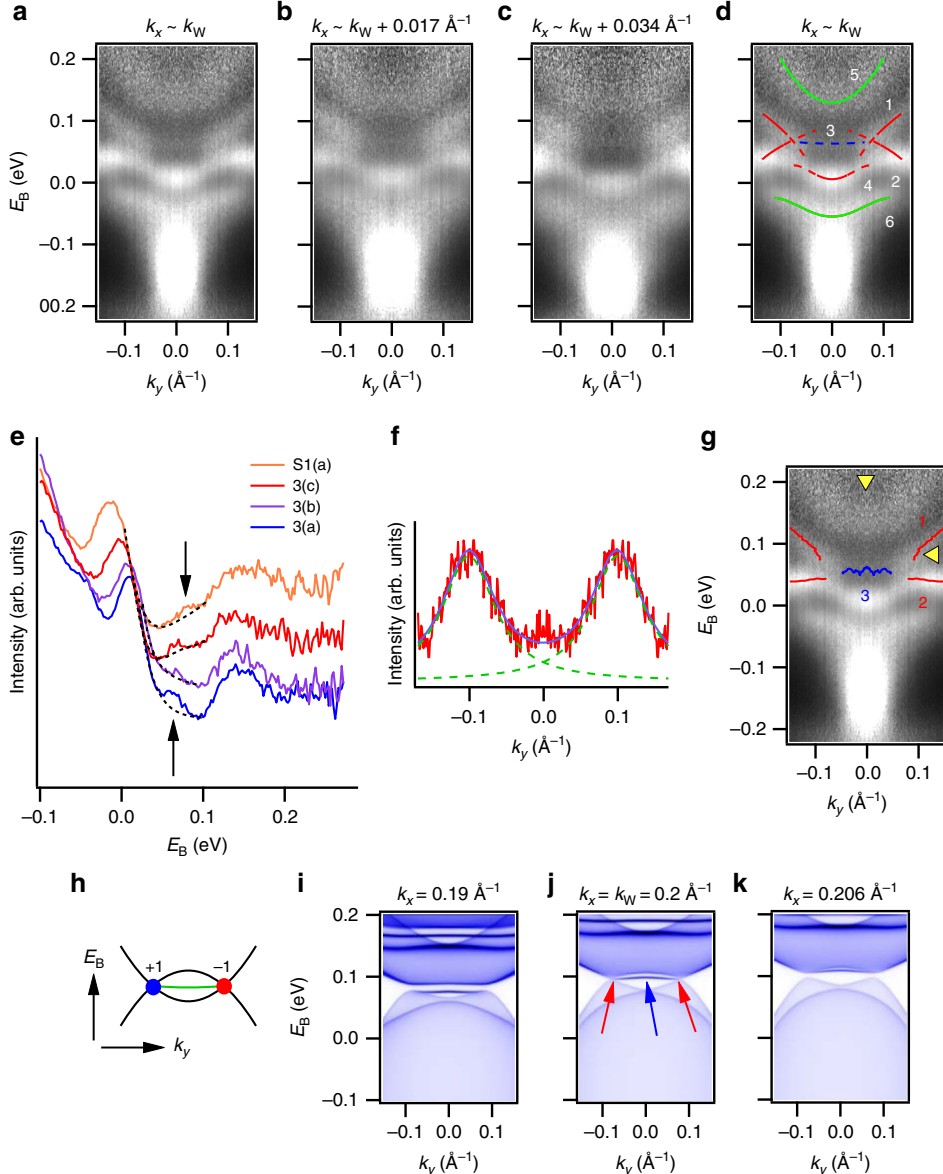

**Fig. 3** Weyl points and Fermi arcs above the Fermi level in TaIrTe₄. **a–c** Pump probe ARPES spectra of TaIrTe₄, showing dispersion above $E_F$ at fixed $k_x$ expected to be near the Weyl points. **d** Same spectrum as **a** but with key features marked. The Weyl cone candidates are labeled 1 and 2, the Fermi arc candidate is labeled 3. **e** Energy distribution curves (*EDCs*) through the Fermi arc at $k_x \sim k_W$, $k_W$ + 0.017 Å⁻¹, $k_W$ + 0.034 Å⁻¹, and $k_W$ + 0.045 Å⁻¹. The *dotted black lines* are fits to the surrounding features, to emphasize the Fermi arc peak, marked by the *black arrows*. We observe signatures of the upward dispersion of the Fermi arc with increasing $k_x$, consistent with ab initio calculations and basic topological theory. **f** An MDC with two large peaks corresponding to the upper Weyl cones. The *dotted green lines* show an excellent fit of the peaks to Lorentzian functions. **g** Same spectrum as **a**, but with key features marked by a quantitative fits to EDCs and MDCs. The *yellow arrows* correspond to the location of the EDCs in **e** and the MDC in **f**. **h** Cartoon of the cones and arc observed in the data, showing what is perhaps the simplest configuration of Weyl points and Fermi arcs that can exist in any Weyl semimetal. **i–k** Ab initio calculation of TaIrTe₄ showing the Weyl points (*red arrows*) and Fermi arc (*blue arrow*). The excellent agreement with calculation suggests that we have observed a Weyl semimetal in TaIrTe₄

above the Fermi level, followed by a 5.92 eV probe laser pulse to perform photoemission[28]. We study $E_B$–$k_x$ cuts near $\bar{\Gamma}$, Fig. 2a–c, with key features marked by guides to the eye in Fig. 2d. Above the Fermi level, we see a crossing-like feature near $E_B \sim 0.15$ eV, labeled 1, and two electron-like bands, 2 and 3, extending out above $E_F$. Below the Fermi level, we observe a general hole-like structure consisting of three bands, labeled 4–6. As we shift $k_y$ off $\bar{\Gamma}$, we find little change in the spectrum, suggesting that the band structure is rather flat along $k_y$ near $\bar{\Gamma}$. However, we can observe that band 4 moves downward in energy and becomes more intense with increasing $k_y$. We find an excellent match between

our ARPES data and ab initio calculation, Fig. 2e–g. Specifically, we identify the same crossing-like feature (green arrow) and top of band 4 (orange arrow). We can also track band 4 in $k_y$ in calculation and we find that the band moves down and becomes brighter as $k_y$ increases, in excellent agreement with the data. The electron-like structure of bands 2 and 3 and the hole-like structure of bands 5 and 6 are also both captured well by the calculation. Crucially, however, we notice a shift in energy between experiment and theory, showing that the sample is hole-doped by ~0.05 eV. Lastly, we plot a constant energy $k_x$–$k_y$ cut at $E_B = E_F$, where we see again that there is little dispersion along $k_y$ near $\bar{\Gamma}$,

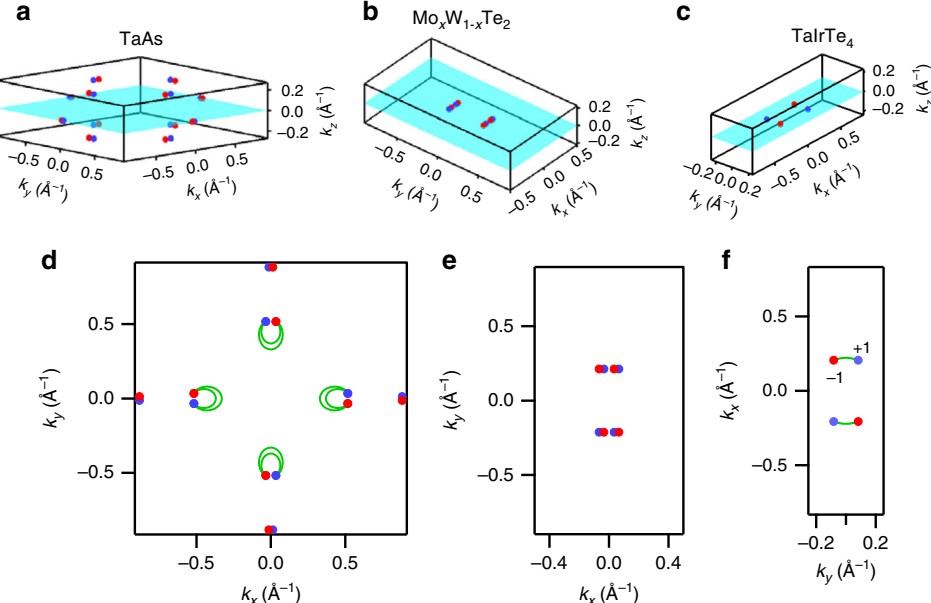

**Fig. 4** Comparison of Weyl point configurations. Weyl points, plotted in *red* and *blue* for opposite chiralities, for **a** TaAs, with 24 Weyl points, **b** $Mo_xW_{1-x}Te_2$, with eight Weyl points and **c** $TaIrTe_4$, with the minimal number, only four Weyl points, making $TaIrTe_4$ a minimal $\mathcal{T}$ invariant Weyl semimetal. The $k_z = 0$ plane is marked in *cyan*. **d**–**f** The projection of the Weyl points on the (001) surface, with topological Fermi arcs. Note that the Weyl points are plotted numerically, while the Fermi arcs are rough cartoons drawn based on ARPES measurements and ab initio results. The *black* frame marks the first Brillouin zone. The length of the Fermi arcs in $TaIrTe_4$ is longer as a fraction of the Brillouin zone as compared to TaAs and $Mo_xW_{1-x}Te_2$

Fig. 2h. We also indicate the locations of the $E_B-k_x$ cuts of Fig. 2 and the $E_B-k_y$ cuts of Fig. 3, to be discussed below. Our pump-probe ARPES measurements allow us to directly measure the electronic structure above $E_F$ in $TaIrTe_4$ and we find excellent match with calculation.

**Evidence for a Weyl semimetal in $TaIrTe_4$.** Now we demonstrate that $TaIrTe_4$ is a Weyl semimetal by directly studying the unoccupied band structure to pinpoint Weyl cones and topological Fermi arcs. Based on calculation, we fix $k_x$ near the expected locations of the Weyl points, $k_x \sim k_W = 0.2 \, \text{Å}^{-1}$ and we study $E_B-k_y$ cuts in ARPES, see Fig. 3a–c, with key features marked by guides to the eye in Fig. 3d. We observe two cone features, labeled 1 and 2, connected by a weak, rather flat arc feature, labeled 3. We find that the cones are most pronounced at $k_x \sim k_W$, but fade for larger $k_x$. Next, we pinpoint the Fermi arc as a small peak directly on the energy distribution curve (EDC) passing through $k_y = 0$, see the blue curve in Fig. 3e, where the dotted black line is a fit to the surrounding features. We further track the arc candidate for $k_x$ moving away from $k_W$ and we find that the arc disperses slightly upwards, by about ~10 meV, see also Supplementary Fig. 1. This dispersion is consistent with a topological Fermi arc, which should connect the Weyl points and sweep upward with increasing $k_x$[29]. We further pinpoint the upper Weyl cone on a momentum distribution curve (MDC) of the $k_x \sim k_W$ cut, Fig. 3f. We find an excellent fit of the Weyl cone peaks to Lorentzians. Using this analysis, we can quantitatively track the dispersions of the Weyl cones and Fermi arc on the $k_x \sim k_W$ cut, Fig. 3g and Supplementary Fig. 2. We note that for the upper Weyl cone we track the bands by Lorentzian fits on the MDC. However, for the Fermi arc and lower Weyl cone, the relatively flat dispersion requires us to track the bands in the EDCs. The EDC peak is challenging to fit, in part because the population distribution is strongly dependent on binding energy for a pump-probe ARPES spectrum. As a result, we track the Fermi arc and lower Weyl

cone through a naive quadratic fit of the band peaks, again see Fig. 3g. We find that the peak trains are nearly linear, see also Supplementary Fig. 2. Based on our pump-probe ARPES spectra and ab initio calculations, we propose that $TaIrTe_4$ hosts two pairs of Weyl points of chiral charge ±1 at $k_x = \pm k_W$, connected by Fermi arcs. This particular structure of two Weyl cones connected by a Fermi arc is arguably the simplest possible, Fig. 3h. We compare our results to calculation in greater detail, Fig. 3i–k. We can easily match the Weyl cones, the Fermi arc and an upper electron-like band, labeled 5 in Fig. 3d. However, we note that from calculation we expect bands 1 and 4 to attach to form a single band, while in our data they appear to be disconnected. We suggest that this discrepancy may arise because photoemission from part of the band is suppressed by low cross-section at the photon energy used in our measurement. In addition, we do not observe good agreement with the lower feature labeled 6 in our calculation, suggesting that this intensity may arise as an artifact of our measurement. At the same time, we consistently observe the broad featureless intensity below the Fermi level in both theory and experiment. Crucially, again we find a mismatch in the Fermi level. In particular, the Weyl points are expected at $E_B \sim 0.1$ eV, but we find the Weyl points at $E_B \sim 0.07$ eV. We note that this sample was grown in a different batch than the sample of Fig. 2 and a comparison with calculation suggests that the second sample is electron doped by ~30 meV, in contrast to a ~50 meV hole doping in the first sample. We propose that the difference in doping of the two samples may arise because they were grown under slightly varying conditions. Lastly, we note that the $k_y$ position of the Weyl points shows excellent agreement in theory and experiment. In summary, we observe an arc which (1) terminates at the locations of two Weyl points; (2) appears where expected in momentum space, based on calculation; and (3) disperses upward with $k_x$, as expected from calculation. The cones (1) are gapless at a specific $k_x \sim k_W$; (2) fade for larger $k_x$; (3) appear where expected, based on calculation; (4) are connected by the arc; (5) show up in pairs only on $k_x \sim k_W$, so that there are

four in the entire Brillouin zone. This provides strong evidence that TaIrTe$_4$ is a minimal Weyl semimetal with four Weyl points.

## Discussion

We compare TaIrTe$_4$ with other Weyl semimetals and consider our results in the context of general topological theory. Weyl semimetals known to date in experiment host a greater number of Weyl points than TaIrTe$_4$. In particular, the well-explored TaAs family of Weyl semimetals hosts 24 Weyl points and Mo$_x$W$_{1-x}$Te$_2$ hosts eight Weyl points[8, 30]. We plot the configuration of Weyl points for TaAs, Mo$_x$W$_{1-x}$Te$_2$, and TaIrTe$_4$, where red and blue circles denote Weyl points of opposite chirality, Fig. 4a–c. It is also interesting to note that the length of the Fermi arc in TaIrTe$_4$ is much longer as a fraction of the Brillouin zone than that of TaAs or Mo$_x$W$_{1-x}$Te$_2$, which can be seen clearly in the projections of the Weyl points on the (001) surface of all three systems, Fig. 4d–f. We see that our discovery of a Weyl semimetal in TaIrTe$_4$ provides the first example of a minimal $\mathcal{I}$ breaking, $\mathcal{T}$ invariant Weyl semimetal. One immediate application of our results is that TaIrTe$_4$ in pump-probe ARPES may provide a platform to observe the time dynamics of carrier relaxation in a Weyl semimetal. More broadly, our results suggest that TaIrTe$_4$ holds promise as a simpler material platform for studying properties of Weyl semimetals in transport and applying them in devices.

## Methods

**Pump-probe angle-resolved photoemission spectroscopy.** Pump-probe ARPES measurements were carried out using a hemispherical electron analyzer and a mode-locked Ti:Sapphire laser system that delivers 1.48 eV pump and 5.92 eV probe pulses at a repetition rate of 250 kHz[28]. The system is state-of-the-start, with a demonstrated energy resolution of 10.5 meV, the highest among any existing femtosecond pump-probe setup to date[31]. The time and energy resolution used in the present measurements were 300 fs and 15 meV, respectively. The spot diameters of the pump and probe lasers at the sample were 250 and 85 μm, respectively. The delay time between the pump and probe pulses was ~106 fs. Measurements were carried out at pressures <5 × 10$^{-11}$ Torr and temperatures ~8 K.

**Single crystal growth and characterization.** For growth of TaIrTe$_4$ single crystals, all the used elements were stored in an argon-filled glovebox with moisture and oxygen levels less than 0.1 ppm and all manipulations were carried out in the glovebox. TaIrTe$_4$ single crystals were synthesized by solid state reaction with the help of Te flux. Ta powder (99.99%), Ir powder (99.999%), and a Te lump (99.999%) with an atomic ratio of Ta/Ir/Te = 1:1:12, purchased from Sigma-Aldrich (Singapore), were loaded in a quartz tube and then flame-sealed under a vacuum of 10$^{-6}$ Torr. The quartz tube was placed in a tube furnace, slowly heated up to 1000 °C and held for 100 h, then allowed to cool to 600 °C at a rate of 0.8 °C h$^{-1}$, and finally allowed to cool down to room temperature. The shiny, needle-shaped TaIrTe$_4$ single crystals, see Supplementary Fig. 3a, were obtained from the product and displayed a layered structure, confirmed by the optical micrograph, Supplementary Fig. 3b, and scanning electron microscope images, Supplementary Fig. 3c. The EDX spectrum displays an atomic ratio Ta:Ir:Te of 1.00:1.13(3):3.89(6), consistent with the composition of TaIrTe$_4$, Supplementary Fig. 3d.

**Ab initio band structure calculations.** We computed electronic structures using the projector augmented wave method[32, 33] as implemented in the VASP[34–36] package within the generalized gradient approximation[37] schemes. Experimental lattice constants were used[38]. A 15 × 7 × 7 Monkhorst-Pack $k$-point mesh was used in the computations with a cutoff energy of 400 eV. The spin-orbit coupling effects were included self-consistently. To calculate the bulk and surface electronic structures, we constructed a first-principles tight-binding model Hamilton, where the tight-binding model matrix elements are calculated by projecting onto the Wannier orbitals[39–41], which use the VASP2WANNIER90 interface[42]. We used Ta $d$, Ir $d$, and Te $p$ orbitals to construct Wannier functions, without performing the procedure for maximizing localization.

**Data availability.** All data is available from the authors on request.

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

## Acknowledgements

I.B. thanks Daixiang Mou and Adam Kaminski for use of their conventional laser ARPES system at Ames Laboratory & Iowa State University during the preliminary phase of this project. I.B. acknowledges the support of the US National Science Foundation GRFP. The work at Princeton is supported by the US National Science Foundation, Division of Materials Research, under Grants No. NSF-DMR-1507585 and No. NSF-DMR-1006492 and by the Gordon and Betty Moore Foundation through the EPIQS program grant GBMF4547 (Hasan). Y.I. is supported by the Japan Society for the Promotion of Science, KAKENHI 26800165. This work is also financially supported by the Singapore National Research Foundation (NRF) under NRF RF Award No. NRF-RF2013-08, MOE Tier 2 MOE2016-T2-1-131 and MOE2016-T2-2-153 (S). T.-R. C. and H.-T. J. are supported by the Ministry of Science and Technology, National Tsing Hua University, National Cheng Kung University, and Academia Sinica, Taiwan. T.-R. C. and H.-T. J. also thank the National Center for High-Performance Computing, the Computer and Information Networking Center of National Taiwan University, and the National Center for Theoretical Sciences, Taiwan for technical support. H.L. acknowledges the Singapore NRF under Award No. NRF-NRFF2013-03.

## Author contributions

The project was conceived by I.B. and P.Y., with guidance from M.Z.H. I.B. carried out the pump-probe ARPES measurements with help from D.S.S. and support from Y.I. P.Y. synthesized and characterized the single crystal $TaIrTe_4$ samples under the supervision of Z.L. Y.I. built the pump-probe ARPES set-up in the laboratory of S.S., with additional support from T.K. T.-R.C. carried out the ab initio calculations with help from G.C. and under the supervision of H.-T.J. and H.L. I.B. analyzed the data and interpreted results with help from D.S.S., T.-R.C., S.S.Z., S.-Y.X., H.Z., G.C., and G.B. All authors contributed to writing the manuscript.

## Additional information

**Competing interests:** The authors declare no competing financial interests.

