## [Peer Review File · Nature Communications]

REVIEWERS' COMMENTS:

Reviewer #2 (Remarks to the Author):

This work is quite compelling and should be of great interest to a broad auditorium of researchers.

Authors properly addressed all of my concerns in the response letter and revised manuscript. Therefore, I would recommend this work for publication in Nature Communications journal.

There are only few minor corrections to be made:

1. Labeling in Fig 2h is hard to see at low zoom levels: Vertical blue lines corresponding to (3a-c, S1a) are so close that they merge into single blue band. Same can be said about Fig. S1 in SI
2. Page 6 refers to non-existent Fig. 3m
3. Caption of S2 refers to non-existent Fig. 3o

Reviewer #3 (Remarks to the Author):

Following my previous comments, the authors have included line cuts of their data, which help to clarify the features shown in Figs 3a-c. Given the agreement between experimental data and ab initio calculations, I believe this manuscript is in principle suitable for publication in Nature Communications.

Re: NCOMMS-17-02201-T, Signatures of a minimal, “hydrogen atom” version of a time-reversal invariant Weyl semimetal, by I. Belopolski, *et al.*

We thank the editors for their consideration of our manuscript and we are excited that our work is likely to be published in *Nature Communications*. We also apologize for the long delay in submitting this final round of revisions. Below we have addressed all remaining concerns of the referees and the editors. We have adjusted the format of our manuscript to adhere to *Nature Communications* guidelines and provided all additional information as necessary.

REPORT OF REVIEWER 2

Reviewer 2: This work is quite compelling and should be of great interest to a broad auditorium of researchers. The authors properly addressed all of my concerns in the response letter and the revised manuscript. Therefore, I would recommend this work for publication in the journal *Nature Communications*.

Authors: We thank the reviewer for her or his time in reading our manuscript and we are thrilled that the reviewer judges our work to merit publication in *Nature Communications*.

Reviewer 2: There are only few minor corrections to be made:

1. The labeling in Fig. 2h is hard to see at low zoom levels: the vertical blue lines corresponding to (3a-c, S1a) are so close that they merge into a single blue band. The same can be said about Fig. S1 in the SI.

Authors: Thank you for the remark. It is fixed.

Reviewer 2: 2. Page 6 refers to non-existent Fig. 3m.

Authors: It is fixed.

Reviewer 2: 3. Caption of S2 refers to non-existent Fig. 3o.

Authors: It is fixed. Thank you again.

REPORT OF REVIEWER 3

Reviewer 3: Following my previous comments, the authors have included line cuts of their data, which help to clarify the features shown in Figs 3a-c. Given the agreement between experimental data and ab initio calculations, I believe this manuscript is in principle suitable for publication in *Nature Communications*.

Authors: We thank the reviewer for her or his remarks and we are excited that the reviewer finds our work potentially suitable for publication in *Nature Communications*.